# A Complex Case of Clino-Syndactyly with Fourth Metacarpal Aplasia

**DOI:** 10.3390/life13091943

**Published:** 2023-09-21

**Authors:** Hermann Nudelman, Aba Lőrincz, Anna Gabriella Lamberti, Tamás Kassai, Gergő Józsa

**Affiliations:** 1Department of Paediatrics, Clinical Complex, Division of Surgery, Traumatology and Otorhinolaryngology, University of Pécs, 7 József Attila Street, 7623 Pécs, Hungary; nuhwaao.pte@tr.pte.hu; 2Department of Thermophysiology, Institute for Translational Medicine, Medical School, University of Pécs, 12 Szigeti Street, 7624 Pécs, Hungary; aba.lorincz@gmail.com; 3Department of Pediatric Traumatology, Péterfy Hospital, Manninger Jenő National Trauma Center, 17 Fiumei Street, 1081 Budapest, Hungary; kassai.tamas@obsi.hu; 4Department of Anatomy, Medical School, University of Pécs, 12 Szigeti Street, 7624 Pécs, Hungary

**Keywords:** clinodactyly, syndactyly, wedge osteotomy, congenital, flaps, plastic surgery, hand surgery

## Abstract

Syndactyly is the most common congenital malformation of the hand, leading to the fusion of the digits and frequently affecting the ring and middle fingers. The incidence is 1 out of 2500 children, predominantly occurring in boys and Caucasians. Clinically, the malformation may present as a soft tissue or bony fusion, resulting in the union of the fingers characterised as complete or incomplete. This fusion may involve the phalanges but may also extend to the carpal/tarsal bones, even to the metacarpal or metatarsal level, rarely to the distal end of the forearm and lower leg. The malformation is mostly isolated but may occur together with other disorders or malformations such as synostosis, acro-syndactyly, cleft hand, clinodactyly, or polydactyly. Syndromic syndactyly can be observed in cases of Apert syndrome, Poland’s syndrome, Pfeiffer syndrome, and many others. A girl born in June of 2019 was diagnosed with congenital malformation of the right hand at birth—affecting the right middle, ring, and little fingers, respectively. After X-ray imaging, the fusion of the third and fourth proximal phalanges to a common metacarpal was identified, forming a unique diagnosis of clino-syndactyly with metacarpal aplasia. Surgical intervention was advocated for, including a wedge osteotomy to correct the synchondrosis at the phalangeal base and a dorsal flap to close the interdigital space created during the correction of the III and IV. fingers. A trapezoid flap for the release of the syndactyly of the IV and V. fingers was applied. The paper aims to present this surgical correction and its results regarding an atypical case of syndactyly with clinodactyly and metacarpal aplasia.

## 1. Introduction

Syndactyly, derived from the Greek words “syn” (meaning “together”) and “dactyly” (meaning “digits”), ranks among the most prevalent congenital malformations affecting the extremities [1,2,3]. It arises due to a developmental separation failure (webbing) during embryological limb formation, resulting in the fusion of the digits. This condition frequently affects the ring and middle fingers. The incidence of syndactyly is estimated to be approximately one occurrence per 2500 children, with a higher prevalence among boys and individuals of Caucasian ethnicity. Mothers aged 40 years or older face an increased risk of giving birth to offspring with congenital limb deformities compared to mothers below 30. Though predominantly a genetic disease, contributing factors such as maternal smoking, lower nutritional or socio-economic status, and increased meat and egg consumption during pregnancy have also been identified [1,2,3,4,5,6]. 

From a clinical perspective, the malformation may manifest as the fusion of soft tissue or bones, leading to the fusion of the fingers characterised as complete or incomplete. This fusion may encompass the phalanges and may also extend to the carpal/tarsal bones, rarely it may extend to the distal end of the forearm or lower leg [1,2,3]. The malformation can be further categorised as simple, complex, and complicated. It can exhibit symmetry or asymmetry and may affect one side (unilateral) or both sides (bilateral), affecting the right, left, or both sides and feet, hands, or both. This condition may present as an isolated malformation (non-syndromic) or as a characteristic feature of hundreds of recognised syndromes. The classification system for non-syndromic syndactyly, initially proposed by Temtamy and McKusick is based on the phenotypic appearance. This classification was extended by Malik et al. in 2012. Currently, non-syndromic syndactyly is classified into nine primary types by the Temtamy and McKusick classification, which are further divided into subtypes [2,3,4]. Non-syndromic syndactyly is predominantly isolated, but it can occur together with other disorders or malformations such as synostosis, acro-syndactyly, cleft hand, clinodactyly, synonychia, or polydactyly. On the other hand, syndromic syndactyly can be observed in specific conditions like Apert syndrome, Poland’s syndrome, Pfeiffer syndrome, and numerous others [1,2,3,4].

Diagnosing the condition relies primarily on patient history and physical examination. Notably, the deformity may be apparent at birth or even detectable on ultrasound images during foetal development. Its treatment typically necessitates surgical intervention as potential benefits outweigh the associated risks. However, surgery is contraindicated in cases of mild syndactyly where functionality is preserved, complex syndactyly with risk of functional deterioration due to surgery, or medical conditions that interfere with the surgical approach [1,4,7]. 

The surgery is ideally performed at around 12 months of age to allow for uninterrupted growth and to minimise the risks of postoperative complications. Similar flaps are employed in both simple and complete syndactyly cases, including the use of rectangular, triangular, omega, and multilobed flaps [8]. There are several other methods and techniques which can be utilised in certain cases, each having its advantages and disadvantages. Surgical release procedures utilising skin grafting encompass a variety of methods with the use of flaps. Alternatively, there are operational methods designed to mitigate the need for grafts. Each technique has its advantages and limitations, with selection based on the case’s specifics and the surgeon’s expertise. In complex cases, surgical planning may involve the utilisation of multiple approaches, potentially necessitating several surgeries and the implementation of multiple flaps, which carry a higher risk for complications. During surgery, a significant emphasis should be placed on meticulously assessing the patient’s neurovascular structures. The course of these structures is often unpredictable, unlike the normal anatomical structure. The amount of osseous involvement directly correlates with the extent of neurovascular involvement and the associated risk of injury [9,10]. 

Clinodactyly refers to the congenital curvature of a digit beyond the metacarpophalangeal (MCP) joint. When the angular deviation is within physiological limits, typically fewer than 5–10°, it is considered a normal variation [11,12]. However, if the coronal angulation exceeds 10°, it is classified as a pathological deformity. This curvature seen in clinodactyly is primarily attributed to the abnormal triangular or trapezoidal shape of one or more phalanges, leading to the misalignment of the interphalangeal joint(s) involved. This malformation causes the affected finger to exhibit an asymmetric longitudinal growth, yielding the curved appearance of the finger. Clinodactyly can occur as syndromic, familial (inherited), or sporadic (occurring by chance). It has been classified by Burke and Flatt into four groups: (1) familial, (2) other congenital anomaly, (3) due to epiphyseal injury, (4) due to triphalangeal thumb. This classification is sufficient to diagnose and find the origin of the malformation; however, it is not suitable for choosing the appropriate treatment option [11,13,14]. Cooney’s classification system differentiates between simple and complex clinodactyly and takes into account the curvature (deviation in degrees) and the involvement of surrounding tissues [12,15,16]. The most recent classification expanded by Ali et al. is based on the severity of angular deviation [11,12,16,17]. The condition most commonly affects the middle phalanx of the fifth digit, as well as the proximal phalanx of the first and second fingers. The aetiology is still not fully understood. It commonly affects males, frequently bilaterally, and has an occurrence rate ranging from 2% to 19% [11,12,17,18]. The malformation has been linked to numerous syndromic anomalies, namely, Klinefelter syndrome, Rubinstein–Taybi syndrome, Fanconi anaemia, Cenani–Lenz syndactyly, and Turner syndrome, and is found in approximately 25% of children with Down syndrome [11]. The association with multiple syndromes highlights the diverse range of conditions where clinodactyly may be observed as a feature. The abnormal growth is attributed to the peculiar configuration of the proximal portion of the phalanx’s epiphyseal growth plate. The epiphyseal plate takes on a C-shaped appearance, yielding incomplete or restricted growth on one side of the phalanx. The presence of a delta phalanx occurs when there is early and complete ossification of the physeal plate, leading to disrupted or restricted growth of the affected digit. In some cases, traumatic injuries of the plate can result in an acquired case of clinodactyly [11,12,17,19].

Its diagnosis is based on a comprehensive physical exam and a detailed review of the patient’s history, which helps differentiate between the congenital or traumatic origins of the deformity [17]. A thorough evaluation should encompass range of motion testing to assess and document the impact on the grasp and pinch functions. On radiographs, a C-shaped physis might be present along with the delta phalanx formation, providing insights into the underlying structural abnormalities [9,11,12,15,19].

Splinting of the affected digit is generally not recommended and has been shown to be ineffective as a treatment option [12,16,19]. Surgical intervention becomes necessary in instances of severe angulation and shortening, especially if the thumb or the radial digits are involved. Such cases could significantly interfere with the hand’s grasp and punch function, justifying the need for surgical correction. The operative technique for clinodactyly is referred to as corrective osteotomy, which is ideally performed at a skeletally mature age to minimise the risk of physeal injury and excessive or inadequate correction [18]. Various methods could be employed, including reverse wedge osteotomy, opening or closing wedge osteotomy, and epiphyseal bracket resection and fat grafting [16,20]. Mild soft tissue deficits that are created can be managed with “Z”-plasty. In contrast, severe soft tissue defects might demand advancement or rotational flaps for adequate coverage and optimal wound healing. Complications include growth arrest, skin deficits, scarring, stiffness, over- or under-correction, and the risk of infection [11,18,20].

Our aim was to present the surgical, therapeutical, and post-operative treatment descriptions for those who might meet a similar case. We aspire to contribute to the existing literature with the details about the case due to the rare and unlikely involvement of the fingers and their comorbidities.

## 2. Materials and Methods

### 2.1. Case Description

A girl born in June of 2019 was diagnosed with congenital malformation of the right hand at birth—affecting the right middle, ring, and little fingers, respectively. Upon physical examination, the III digit deviated from its origin at the metacarpophalangeal joint (MCP) radially by approximately 45°, while the IV finger deviated to the ulnar side by 45° from the MCP joint. Clinodactyly is apparent when looking at the proximal interphalangeal (PIP) joints of the III and IV fingers, as they are positioned in a close to 90° deviation. Between the IV and V fingers webbing was visible, resembling a plate, adding to the syndactyly component of the malformation (Figure 1A). The fingers’ PIP and distal interphalangeal (DIP) joints were movable; however, passive flexion and extension were restricted. The passive flexion of the MCP was close to 90°, while extension was limited. Radiological evaluation revealed a common metacarpal of the III and IV fingers (Figure 1B). Altogether, there were only four metacarpal bones in the patient’s hand. The common MCP joint was fused with the proximal phalanxes of the middle and ring fingers. The plate acquired a deviated position which led to the angular displacement of the fingers. The PIP joints’ clinodactyly was apparent as the large degree of deviation speaks for itself. In this case, the middle finger’s deviation was a close 90° deviation towards the ulnar side, while the ring finger deviated approximately 40° radially with the interdigital webbing of the little and ring fingers (Figure 1C). Functionality was restricted both actively and passively, with some flexion present. Grasp and pinch functions were decreased as the child could not hold items in their palm faced down. When grabbing items, the child used the index finger and thumb to hold onto the object. These findings formed the diagnosis of clino-syndactyly of the III and IV fingers’ base and PIP joints, with incomplete cutaneous syndactyly of the IV and V digits, paired with metacarpal aplasia of the IV metacarpal bone.

### 2.2. Surgical Method

Surgical intervention was advocated for, which included a wedge osteotomy to correct the synchondrosis at the phalangeal base and a dorsal flap to close the interdigital space that was created during the correction of the III and IV fingers. A trapezoid flap was applied for the release of the syndactyly of the IV and V fingers. The child was aged three at the time of the surgery. Clinical application of the technique was accepted and permitted by our medical review board, the Hungarian Pediatric Trauma Committee, and the Hungarian Pediatric Surgery Committee. The work was performed in Pécs, at the Surgical Division, Department of Paediatrics, Medical School, University of Pécs, 7 József Attila Street, H7623 Pécs, Hungary.

Prior to undergoing general anaesthesia, the patient received antibiotic prophylaxis. The operation took place in exsanguinated conditions, which totalled 110 min, with the patient in a supine position. The patient received postoperative anticoagulant therapy, as is used routinely. 

After disinfection and draping, the incision lines on the palmar and dorsal surface of the right hand were marked (Figure 2). As the first step, we aspired to solve the clino-syndactyly of the III and IV fingers’ base with wedge osteotomy and the utilisation of a dorsal flap, after which we focused on the webbing between the IV and V fingers. The latter was repaired with the application of trapezoidal island flaps. 

We followed the incision lines based on the designed Dorsal W-Z flap, first dorsally, then palmarly. As a first step, the digital artery and nerve (Figure 3), which was positioned quite distally, were identified. Proximally, at the base of the proximal phalanges, the synchondrosis became visible (Figure 3). After marking the appropriate position of the osteotomy, a part of the synchondrosis together with a small piece of the metaphysis was removed with the help of an oscillating saw without coming into contact with the growth plates of the proximal phalanges, thereby performing a closed wedge osteotomy. With the help of a Luer, we deepened the area after which we could rotate the fingers medially, towards each other. Upon volar approach, we noted that the branching of the digital nerve took place rather distally. With the help of micro-equipment, we separated the common digital nerve so its branching was positioned more proximally in order to aid the turning of the fingers into the ideal position. We lead a strong 2/0 Vicryl^®^ absorbable thread subperiosteally around the bone in order to bring the III and IV radii closer to each other. Before the final fixation, we applied the dorsal flap, which was meant to cover the interdigital space between the fingers. Subsequently, we tied the strong 2/0 (Vicryl^®^) thread so that the two proximal phalanges were positioned parallel to each other. After confirming the adequate position of the fingers, the remaining soft tissue defects were covered by the flaps, yielding no deficits of any kind. We sutured the area with a 6/0 (Vicryl^®^) absorbable thread with the application of interrupted and continuous sutures. 

At this point, the process of exsanguination was temporarily halted to check the circulation of the fingers, which was ideal. All surgeons present agreed to correct the IV. and V. fingers’ interdigital webbing to ease the syndactyly affecting the ring and little fingers as well. To achieve this, incision lines were marked in the shape of two trapezoid flaps, facing each other between the fingers. One was marked on the palmar side, while the other was positioned dorsally. This created a soft tissue defect, approximately 2 × 0.5 cm, over the ring finger’s proximal phalanx which was covered with a full-thickness skin graft obtained from the excess skin of the little finger. After making sure that the interdigital fold was well-positioned and had acquired a deep enough orientation, the flaps were sutured with a 6/0 thread and a continuous suture technique. We applied Bacitracin-containing swathing and bandaging, after which the exsanguination was terminated, permanently this time. We checked for capillary reflex and proper microcirculation when we noticed that the ring finger differed in colour from the rest of the fingers. It was whitish, therefore we removed and reapplied new swathing and bandages in a looser fashion. Following this, the microcirculation of the finger returned to normal and signs of arterial congestion ceased. The patient received a splint and dressings. The dressings were changed frequently and were utilised until the 2nd week of the post-op period.

### 2.3. Post-Operative Therapy and Follow-Up

Post-op antibiotic prophylaxis was given in the form of Augmentin (Amoxicillin + Clavulanic Acid) syrup 3 × 4 mL for seven days. During the early post-operative period, there were no signs of circulatory anomalies, and the microcirculation was continuous. The control radiograph described the post-osteotomy state as the fusion of the proximal phalanges’ proximal epiphyses, forming a Y shape from a common metacarpal (Figure 4B). One day after surgery, during rebandaging, proper capillary reflex and microcirculation were noted, with non-reactive scarring (Figure 4). On the 3rd day after surgery, the patient was discharged to their home for recovery. On control examination, ten days after surgery, circulation was adequate with proper capillary reflex. On the dorsal surface of the hand, a slight hematoma was visible, with mild swelling and non-reactive scars. Rebandaging and Bacitracin-containing swathing were applied. Three days later, the bandaging was removed, and the wound was left to heal on its own two weeks after surgery. Further control examinations on days 21 and 35, corresponding to weeks three and five, noted the proper depth of the IV. interdigital space and the adequate circulation of the fingers. The scars remained non-reactive and continued to heal while the swelling went down (Figure 5). The fingers were straight; however, functionality could not be tested yet. 

During the postoperative period, the patient underwent physiotherapeutic training and daily exercise to better the functionality of the fingers. After immobilisation, physiotherapy began one week after surgery. This included passive range of motion therapy and the stretching of unaffected digits for five minutes in two-hour intervals as movement is essential for adequate blood flow and healing. Daily activities were limited for the first two weeks with splinting, although movement of the fingers was encouraged. Exercises continued for the next four weeks with a gradual ease of restrictions for daily activities in the form of occasional splint removal. At week 6, active range of motion exercises were introduced; however, some restrictions still applied to activities of daily living. After the 8th week, strengthening exercises were started and functional activities were gradually reintroduced. At this point, immobilisation was only required during activities that put excessive stress on the affected fingers. During the control examination on week 8, it was noted that the wound remained non-reactive, and the swelling had completely disappeared (Figure 5). After six months (24 weeks), the patient was aged 4, and there was no recurrence of deformity and the wound healed with minimal scarring (Figure 5). The enhanced functionality of the fingers was described by the parents. Upon examination, the grasp and pinch functions were identical to the other side and were not lacking in strength, coordination, or range of motion. The PIP joints of the III and IV fingers were still restricted in motion; however, their orientation was now ideal to allow for a proper grasp. 

## 3. Discussion

The fact that a metacarpal was absent stirred concern regarding the long-term growth of the digit, and it made us question whether its functionality will continue to last. This case was one of a kind in this aspect, as the bones could not be separated into two metacarpals. Surgical correction of complex syndactyly includes the Flatt technique, the dorsal omega flap, the M-V flap, the reverse W-M flap, and the combination of the V-Y and rectangular flap with full-thickness grafts. These can be obtained from various areas, such as the groin (frequently at the region of the anterior superior iliac spine), the dorsal or lateral thigh, the medial upper arm, the distal wrist, or the dorsal metacarpal region. In our case, we applied an individually made dorsal flap design to correct the III and IV fingers’ syndactyly. This approach entailed a palmar incision line resembling that of a Z-plasty and a dorsal flap, composed of two bilobed triangular flaps on each side with a dorsal rectangular flap in the middle, collectively forming a shape resembling a W (Figure 2). For the correction of the component responsible for the IV and V fingers’ involvement, two trapezoid flaps were used, which were also individually tailored for this patient during surgical planning. The greatest advantage of our technique was that it enabled the surgeons to operate without the need for skin grafting from additional donor sites. The release was successful and besides the soft tissue deficit that was created during the first incisions which formed the flaps, the operation was unremarkable. We decided on the local graft instead of a rotational flap as the turning of the lobe would have placed the skin under tension in a way not suitable for healing. To avoid this, we utilised the excess skin and grafted a piece onto the IV. finger’s inner side, facing the interdigital space. 

Complications of surgical correction include web creep, graft failure or maceration, infection, joint contractures, and scar hypertrophy [1]. Factors that play a role in the formation of web creep after surgery include complex syndactyly, wound dehiscence, secondary healing due to graft failure, the use of split-thickness skin graft, and poor flap design. Graft failure or maceration is often associated with active children and improper immobilisation and, if substantial, requires a secondary grafting procedure. Joint contractures which are due to the contracture of scars on the palmar aspect of the interphalangeal joints can be corrected by Z-plasty or skin grafting [1,7,8]. 

Clinodactyly, as mentioned, can be corrected by open or closed wedge osteotomy, reverse wedge osteotomy, and Vicker’s physiolysis [11,12,18]. All methods have their advantages and disadvantages; however, in this case, we applied closed wedge osteotomy. This is due to the fact that the base of the fingers was fused, forming a synchondrosis. As there was a single metacarpal connecting to both proximal phalanges, the removal of a wedge was necessary for achieving proper positioning. Both reverse wedge and open wedge methods were unfit for this particular case as the open wedge technique would have required a more aggressive operative approach, while reverse wedge osteotomy was deemed unfit as it would have required two wedges in order to turn both fingers appropriately. These would have required more extensive open surgery and hence, would have led to more scar formation. Complications include loss of motion, recurrence of deformity, infection, non-union, or joint stiffness [11,16,17]. The unlikely location of the curvature called for a unique approach to preserve the functionality of the III and IV fingers. The orientation at the joint mandated careful repositioning to avoid loss of function or loss in the range of motion. By removing the wedge-shaped fragment from the fused joints encompassing a bit of the metaphysis, we enabled the rotation of the base of the digits into the appropriate position. We cut and realigned the base of the fingers so that the growth plates were not affected to preserve the lengthening of the fingers during the growth period. If the wedge is too deep or not deep enough, the lengthening of the fingers as well as their functionality could be affected; herein lies the greatest disadvantage of the technique. Thanks to the expertise of the surgeons, the shape, depth, and location of the osteotomy was more than ideal. The main advantage of using absorbable threads and turning the fingers and fixating them is that they require no removal and lead to minimal soft tissue irritation. Immobilisation is essential in the first week of the postoperative period. As the threads do not provide such stability to the affected digits, there is a risk of implant failure and thus, splinting of the affected digits is an absolute necessity. 

The patient was aged three at the time of the surgery, and two years post-op, at age five, the child presents good functionality without complications or scarring. The growth of the fingers is symmetrical, and the deformity did not reoccur during growth. There were no signs of web creep affecting either the III and IV or the IV and V interdigital space. 

## 4. Conclusions

Firstly, we wanted to show that even after age one, in a complicated case, syndactyly could be corrected. The patient was aged three at the time of the surgery, with two subsequent malformations affecting the right hand. Regardless, the surgery was successful, and even though one metacarpal was amiss, we managed to correct the curvature of the fingers while salvaging vital arteries and nerves and bettering the functionality of the affected hand. 

The malformation formed a rare orientation at the MCP joint, which was challenging to correct as it required pre-op planning and careful execution during surgery. We hope to contribute with this report to the correction of inborn hand deformities. Furthermore, the surgical, therapeutical, and post-operative treatment descriptions could be of high value to those who meet a similar case; to them, we dedicate this report.

## Figures and Tables

**Figure 1 life-13-01943-f001:**
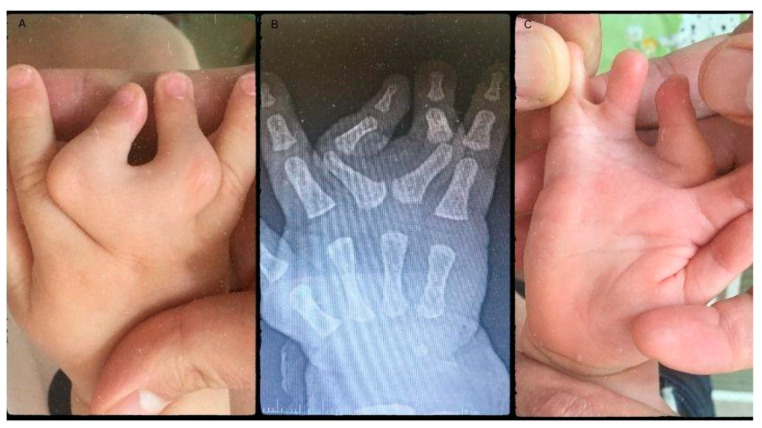
Pre-operative pictures depicting the malformations (**A**,**C**). The radiograph shows the degree of deviation and the missing metacarpal (**B**).

**Figure 2 life-13-01943-f002:**
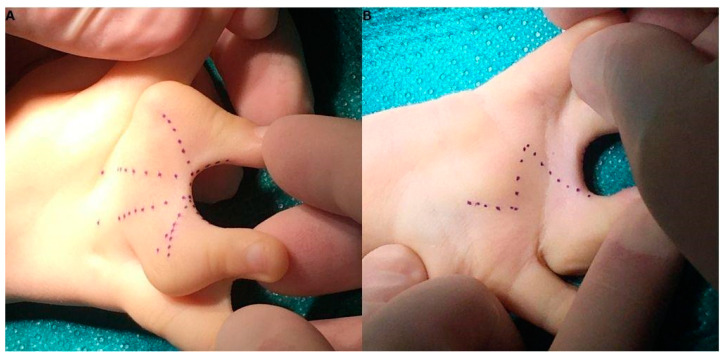
Incision lines can be seen in a W-pattern, combining a rectangular flap and a Z-plasty method between the III and IV fingers dorsal aspect (**A**), and a Z-pattern on the palmar side (**B**).

**Figure 3 life-13-01943-f003:**
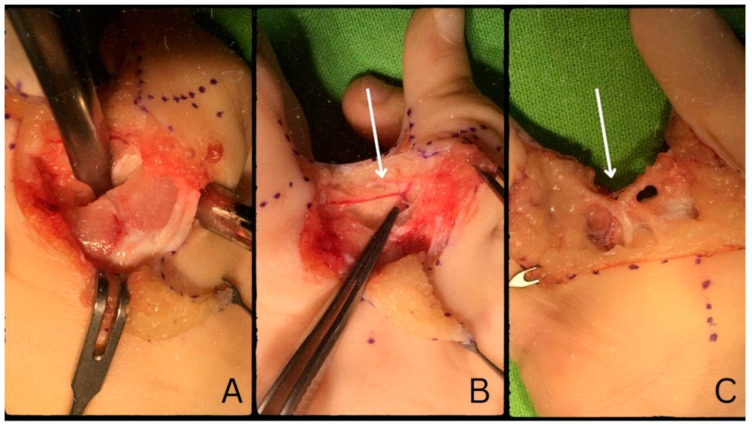
Synchondrosis at the base of the proximal phalanges (**A**). Digital artery (**B**). Common digital nerve prior to separation (**C**).

**Figure 4 life-13-01943-f004:**
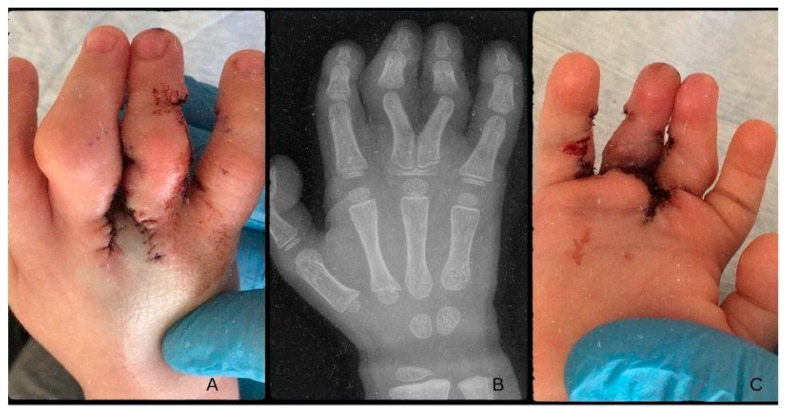
Post-operative images one day after surgery, dorsal side (**A**) and palmar side (**C**). The radiograph describes the postoperative state six weeks after the surgery (**B**).

**Figure 5 life-13-01943-f005:**
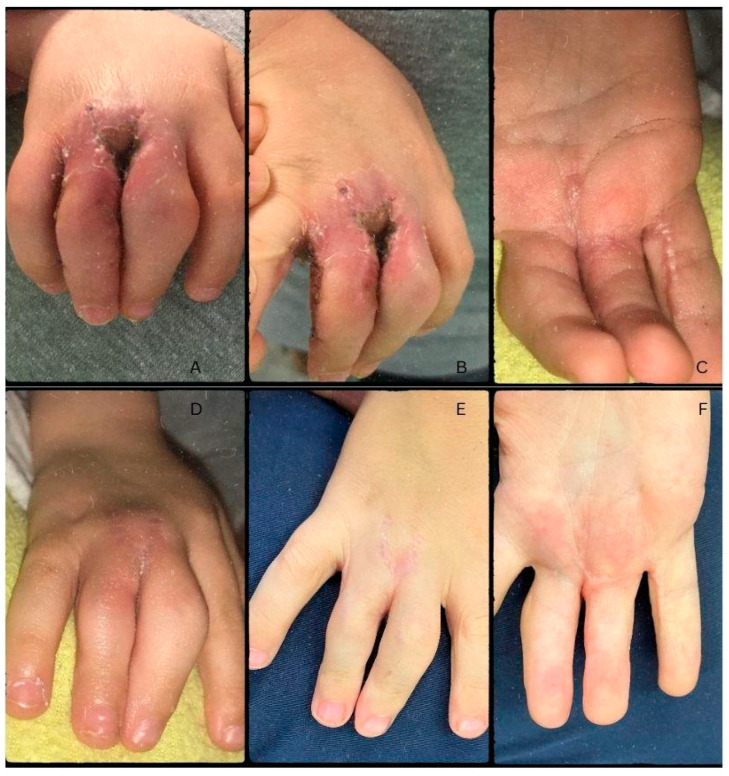
Images taken five weeks after surgery (**A**,**B**). The healing is almost complete eight weeks post-op (**C**,**D**). Long-term follow-up images at 24 weeks (**E**,**F**).

## Data Availability

The data are contained within this article.

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
