# Peer review of "A Complex Case of Clino-Syndactyly with Fourth Metacarpal Aplasia"

_life, 2023, doi:10.3390/life13091943_

Round 1

Reviewer 1 Report

Introduction:

While well-written and detailed, I feel it may be overly long and complex for a journal article. As this is not a book chapter, it would be best to focus on the key points and pique the interest of the average reader. If they wish to delve deeper, they can refer to the relevant references.

Avoid repetitions.

I would suggest ending the introduction by clearly stating the purpose and importance of the current study. Specifically, the final paragraph could summarize:

-The objective/aim of this particular research

-How this work builds on previous findings in the field

-The unique contribution this study makes to the existing literature.

-Why this research question/topic is significant and worth investigating.

Surgical Method:

Figure 3: I suggest adding arrows or asterisks to highlight key features/areas of interest.

What was the age of the patient when she underwent surgery? Please specify the age in the case report. Adding this detail would provide important clinical context.

Please include the patient's age at each follow-up visit. Stating the age at the time of surgery and during subsequent appointments will allow readers to better contextualize the case progression.

Line 252, 315, 360: “Figure X” please correct. 

Figure 4: “Figure 4: Post-operative images one day after surgery, dorsal side (A) and palmar side (C). 284 Radiograph describes the postoperative state … weeks after the surgery (B).”  how many weeks?

Please specify the type of postoperative immobilization used - was it a plaster cast, splint, or just dressings?

Discussion:

The discussion section contains some repetitiveness from the introduction. Please avoid restating background information and instead focus on analyzing the current case. Specifically, consider:

-How this case differs from previous examples in the literature.

-The advantages and disadvantages of the approach used.

There is no need to exhaustively detail all possibilities, just relate to the context of this patient. Additionally, compare and contrast unique aspects of this case to build new insights.

Line 369: figure Y ?

Author Response

Dear Reviewer 1,

We thank you for your observations. Please see the attachment.
Sincerely,
H.N and A.L

Reviewer 2 Report

Interesting case report on a rare condtion. Some modifications are needed to make the manuscript easier to read:

Line 9: affiliation number 3 is missing

Introduction is quite long; I would particularly shorten art from line 70-103 and 146-171 as specific surgical technique is described later

Line 210 when did the surgery took place, how old was the patient? And why at that timing? (maybe discuss this last point in discussion)

Line 252 Figure X (??) is missing

Line 285 in didascaly of Figure 4 timing of radiography is not clear, and if different from when the clinical pictures were taken it might be presented as a separated Figure.

Line 287 specify which kind of physiotherapeutic training

Line 316 another Figure X (??) is missing

Line 360 and 370 Figure X and Figure Y ???

In discussion many lines just repeat what has already been described in introduction, avoid this (maybe you can erase some more line in the introduction)

Author Response

Dear Reviewer 2,

We thank you for the important notes and observations. Please see the attachment.

Sincerely,
H.N and A.L.

Round 2

Reviewer 1 Report

-